# Latent Profile Analysis of Suicidal Ideation in Chinese Individuals with Bipolar Disorder

**DOI:** 10.3390/bs14050360

**Published:** 2024-04-25

**Authors:** Yanmeng Pan, Huaizhi Wang, Yimeng Geng, Jianbo Lai, Shaohua Hu

**Affiliations:** 1Department of Psychiatry, The First Affiliated Hospital, Zhejiang University School of Medicine, Hangzhou 310003, China; 12018602@zju.edu.cn (Y.P.); dannywang1104@foxmail.com (H.W.); 12118462@zju.edu.cn (Y.G.); 2School of Psychiatry, Wenzhou Medical University, Wenzhou 325035, China; 3Nanhu Brain-Computer Interface Institute, Hangzhou 310003, China; 4Zhejiang Key Laboratory of Precision Psychiatry, Hangzhou 310003, China; 5Brain Research Institute, Zhejiang University, Hangzhou 310003, China; 6Zhejiang Engineering Center for Mathematical Mental Health, Hangzhou 310003, China; 7MOE Frontier Science Center for Brain Science and Brain-machine Integration, Zhejiang University, Hangzhou 310003, China; 8Department of Psychology and Behavioral Sciences, Zhejiang University, Hangzhou 310003, China

**Keywords:** bipolar disorder, latent profile analysis, suicidal ideation

## Abstract

Individuals with bipolar disorder (BD) have a greater suicide risk than the general population. In this study, we employed latent profile analysis (LPA) to explore whether Chinese individuals with different phases of BD differed at the levels of suicidal ideation. We recruited 517 patients. Depressive symptoms were measured using the 24-item Hamilton Depression Rating Scale (HAMD-24), and manic symptoms were evaluated using the Young Mania Rating Scale (YMRS). The extent of suicidal thoughts was determined through the Beck Scale for Suicide Ideation (BSSI). The scores of HAMD and YMRS were used to perform LPA. LPA categorized participants into three classes: one exhibiting severe depressive and mild manic symptomatology, another showing severe depressive and severe manic symptomatology, and the third one displaying severe depressive and intermediate manic symptomatology. Suicidal ideation levels were found to be remarkably elevated across all three classes. Additionally, the three classes showed no significant differences in terms of suicidal ideation. Our research confirms the link between depressive symptoms and suicide, independent of the manic symptoms. These findings carry meaning as they provide insight into the suicide risk profiles within different phases of BD.

## 1. Introduction

Bipolar disorder (BD), also called manic–depressive illness, is a severe mental disorder often linked to premature mortality. Many studies have revealed that patients with BD have a remaining life expectancy 10–20 years shorter relative to healthy controls [1,2,3]. Unmistakably, the literature consistently documented a notably higher rate of unnatural deaths (especially suicide) contributing to excess mortality [4,5,6]. The suicide rate in individuals with BD is approximately 20–30 times higher compared to the general population [7,8]. Roughly 34% of those diagnosed with BD have made suicide attempts [9]. Patients with BD have a greater likelihood of dying by suicide than those with major depressive disorders [10,11,12]. Several factors (e.g., depressive or mixed symptoms, severity of symptoms, anxiety and other psychiatric condition comorbidities) may increase the risk of suicide in BD patients [7,10,13].

It remains unclear how the incidence of suicide varies during different phases of BD. Retrospective and cross-sectional studies have frequently linked mixed symptoms, which involve the interaction of depressive and manic symptoms, to the highest levels of suicidal ideation (SI) [14,15,16]. The relationship between mixed episodes (or ever mixed states) and suicidal behavior is more pronounced due to the presence of psychomotor agitation and racing thoughts, which are considered independent predictors of suicidal ideation [17,18]. Depressive symptoms, but not manic symptoms or mixed symptoms, were strongly correlated with suicidal thoughts and actions, as per other studies [7,19]. The presence of mixed symptoms or the combination of manic symptoms do not suggest a higher suicide risk beyond what is already indicated by the depressive symptoms alone [20]. Hence, it is imperative to examine how particular symptoms influence suicidality in BD.

Suicidal ideation and behavior across various mood states is often neglected in previous studies. And, most studies define the state of BD by applying a specific cut-off value, neglecting the individual heterogeneity. Latent profile analysis (LPA) is different from variable-centered analysis. LPA is a person-centered analysis, which uses maximum likelihood estimations to find patterns of multiple variables within individuals but not the effects of individual variables [21,22,23]. This method enables the division of individuals into smaller and more similar subgroups. 

The current study seeks to employ LPA to identify subgroups of Chinese individuals with BD and investigate the association between these groups and suicide ideation.

## 2. Materials and Methods

### 2.1. Participants

This was a single-center cross-sectional study. This study is part of the Chinese Longitudinal and Systematic Study of Bipolar Disorder (CLASS-BD), with the ClinicalTrials registration number of NCT05480150. This study was approved by the Ethics Committee of the First Affiliated Hospital, School of Medicine, Zhejiang University (IIT20210291B-R1). Participants for this study were recruited from the Department of Psychiatry, the First Affiliated Hospital, Zhejiang University School of Medicine, from March 2021 to December 2023. Once the participants consented to participate, they were screened by two trained psychiatrists fulfilling the Chinese version of the Mini-International Neuropsychiatric Interview (M.I.N.I.) via a systematic clinical interview. To be included, all participants needed to meet the Diagnostic and Statistical Manual of Mental Disorders, 5th Edition (*DSM-5*), criteria for BD. Eligible participants in this study underwent an assessment conducted by trained personnel using the 24-item Hamilton Depression Rating Scale (HAMD-24) and the Young Mania Rating Scale (YMRS). Exclusion criteria included (i) chronic and severe physical disease comorbidities (e.g., cardiovascular diseases and tumors) and (ii) a history of primary substance abuse, mood-incongruent psychotic symptoms and schizophrenia spectrum disorder.

### 2.2. Psychometric Tools

#### 2.2.1. Screener: M.I.N.I.

The Mini-International Neuropsychiatric Interview is a concise and valid structured diagnostic interview with an administration time of around 15 min [24]. According to Chinese Guidelines for the Prevention and Management of Bipolar Disorders (3rd version), the M.I.N.I. is mainly used to screen and diagnose sixteen kinds of mental disorders (including BD) and one kind of personality disorder. The M.I.N.I. has been proved to have robust reliability and validity and is used globally [25,26,27].

#### 2.2.2. Profile Indicator 1: HAMD-24

The Hamilton Depression Rating Scale we utilized in this study is a 24-item clinician-administered rating scale that assesses the symptoms of depression over the past week [28,29]. The majority of items are rated from 0 (no symptoms) to 4 (severe symptoms). The total scores range from 0 to 76. The HAMD-24 ≤ 8 indicates no depressive symptoms [30].

#### 2.2.3. Profile Indicator 2: YMRS

The Young Mania Rating Scale is an 11-item examiner rating scale that evaluates hypomanic and manic symptoms in the last week [31]. The scale for most items goes from 0 to 4, except the 5th, 6th, 8th and 9th, which range from 0 to 8. The total scores are from 0 to 60. A YMRS value ≤ 6 indicates no manic symptoms [32].

#### 2.2.4. Outcome: BSSI

The Beck Scale for Suicide Ideation is the one of the most widely used self-report scales that measures the severity of SI [33,34]. The BSSI is a three-point Likert scale including 19 items. Each item on this scale is scored from 0 to 2, giving a total score that ranges from 0 to 38. The higher the score, the more severe the suicide ideation. The best cut-off value to indicate a high risk of SI is BSS ≥ 3 [35]. The BSSI can screen for current suicidal ideation (SI-C) and can also serve as an exploration of the severity of suicidal thoughts at the worst status (SI-W). The BSSI is widely utilized in BD due to its demonstrated reliability and validity [33].

### 2.3. Statistical Analysis

The primary analyses were carried out using LPA. LPA is an extension of latent class analyses of continuous observed variables, which assumes that the observed responses patterns in a population can be explained by an underlying categorical factor. There are several objective criteria that assist in determining the goodness of fit and selecting the final model. Fitting the model requires estimating log-likelihood, information criteria (e.g., Akaike information criterion (AIC), Bayesian information criterion (BIC) and sample size-adjusted BIC (aBIC)), entropy, the Lo–Mendell–Rubin (LMR)-adjusted likelihood ratio test, the bootstrapped parametric likelihood ratio test (BLRT), as well as the proportion of the population in each class [36]. The information criteria (AIC, BIC and aBIC) were first mentioned in our report to protect against data overfitting in models. A smaller IC indicates a better fit for the model [37,38]. Next, we presented two likelihood ratio tests that compared a k-class model to a k − 1 class model, which indicated no improvement by a non-significant p value [39]. Finally, entropy (which ranges from 0 to 1) is an indicator of classification certainty. The higher the entropy, the higher the classification certainty [40]. The selected optimal classification should have as a small number of profiles as possible while achieving an acceptable model fit. All profiles should include at least 5% of the sample and should be interpretable [41]. LPA was conducted with Version 8.3 of Mplus. 

Once the final model was determined, the most probable latent profile membership was exported to SPSS (Version 26.0). This was conducted to examine the differences in clinical characteristics among the demonstrated classes. A comparison of the categorical variables was carried using the Chi-squared test. One-way analysis of variance (one-way ANOVA) and *t*-tests were conducted to test the continuous variables. The demographic factors included current age, gender, height, weight, marriage (yes/no), education, family history, first episode (depression, mania/hypomania and mix) and years since the first onset of symptoms. Afterward, to examine the correlates of SI among the classes, suitable one-way ANOVA was used to compare the mean levels across the profiles. If we detected significant intergroup differences, appropriate pairwise post hoc comparisons were performed. Then, we assessed the interaction effect between classes and clinical characteristics by adding gender as a covariate. 

## 3. Results

### 3.1. Latent Profile Analysis: Number of Divided Classes

Table 1 provides a summary of the model fit indices. The entropy values of the two-class and three-class models exceeded 0.8, indicating over 90% accuracy in individual classifications. The best model was determined based on the AIC, BIC, aBIC, LMR and BLRT. When taking into account all the model selection criteria, the three-class model emerged as the superior fit compared to the two-class model, and it was also more interpretable. 

### 3.2. Demographic and Clinical Profiles

Table 2 provides a breakdown of the demographic and clinical characteristics for each of the three classes. This study involved a total of 517 participants. The age distribution varied from 10 to 55 years (M = 22.10, SD = 7.61). Out of the total participants, more than half were female (n = 354; 68.47%) and 163 (31.53%) were male. Over half of the individuals in the sample who provided their marital status were not married (n = 422; 81.5%). Educationally, about two-thirds (n = 348; 67.31%) had equal to a ninth-grade education and more than one-third (n = 200; 38.68%) had received education from a college or vocational school. Regarding family history, 319 of 517 (83.95%) answered “no” to the question “Have any of your blood relatives (e.g., grandparents, parents, children and so on) had bipolar disorder or other psychiatry disorders”. Only 63 of 517 answered “yes” to the question and reported a detailed family history. Over half of the patients were depressive at the first onset. There were no significant differences in demographic characteristics across the three groups.

### 3.3. Clinical Details of the Three Classes

Figure 1 summarizes the average scores of the HAMD and YMRS. 

The first class showed high scores on the HAMD items but exhibited low scores on the YMRS items. The HAMD score of this group surpassed the cut-off score (M = 23.14; SD = 9.69). And, the YMRS score indicated a low likelihood of experiencing mania or hypomania (M = 3.93; SD = 2.75). The first class was labeled the mild mania–depression group (n = 382). 

The second class showed the highest average HAMD score (M = 32.23; SD = 10.22) and YMRS score (M = 23.43; SD = 3.38), which were respectively within the realm of severe depression and obvious mania. The second class was subsequently referred to as the severe mania–depression group (n = 30).

The third class exhibited a high level of the HAMD score (M = 24.74; SD = 7.35) and a moderate level of the YMRS score (M = 13.72; SD = 2.55). Therefore, the third class was designated the moderate mania–depression group (n = 105).

### 3.4. Contrasts in SI among the Three Classes

As shown in Figure 2, the mean scores of SI-C and SI-W in all three class were all above the cut-off value (≥3). Regardless of the current or worst situation, the severe mania–depression group consistently had the lowest average scores on SI. The analysis of variance found no significant differences among the three classes.

As shown in Figure 3, the gender factor had a notable influence on the BSSI. Women exhibited a considerably greater risk of SI in comparison to men. The results of the covariance analysis were as follows (Table 3). Nonetheless, gender did not have a statistically significant moderating effect.

## 4. Discussion

The main objectives of this study were to identify the variation in suicide risk among Chinese patient subgroups in BD based on depressive and manic/hypomanic symptoms. Building upon previous studies, this study employed LPA to examine BD, specially focusing on the two core symptoms (mania and depression) [42,43]. There were two main findings. Firstly, through LPA classification, three classes of BD were identified (the mild mania–depression group, the severe mania–depression group and the moderate mania–depression group). All three classes presented significant depressive symptoms, and the only difference was the severity of the mania. Additionally, all three classes had higher scores of SI than the general population [44,45]. Secondly, the difference in mania levels across the groups did not correspond with SI no matter at the current state or at the worst state. Thirdly, our study also uncovered significant differences in gender.

### 4.1. The Association between Suicide Risk and Different Classes

We confirmed findings that patients with BD are at a high risk of suicide. Mixed symptoms (depressive symptoms present during mania and vice versa) do not exacerbate or alleviate the risk of SI beyond that attributable to depressive symptoms. The primary predictor of suicide risk in BD is the severe levels of depression, regardless of the presence of manic symptoms. Memory biases may contribute to the heightened suicide risk associated with depression. Memory biases are viewed as crucial cognitive processes underlying depressive symptoms and are strongly related to depression [46]. Depressed individuals tend to have an enhanced memory for negative events, including thoughts of suicide [47].

Another potential interpretation might be that manic impulsivity may elevate suicide attempt risks without affecting suicidal thoughts [48,49]. However, the rates of attempting suicide were not evaluated in this study. This could also be explained partly by the common characteristics that depressive symptoms and mixed symptoms share, such as sociodemographic, biological and psychological components [50]. An additional investigation is necessary to clarify the potential mechanisms linking depressive symptoms to suicidality in specific groups.

### 4.2. Clinical Significance

According to this study, individuals with BD experiencing a depressive episode or mixed symptoms may be at an equally high risk of SI, regardless of the severe levels of mania. The presence of depressive symptoms can heighten the risk of suicide in individuals diagnosed with other psychiatric disorders, including anxiety disorders. BD presents a unique risk profile. Thus, it is essential to regularly access the symptoms of mood disorders and evaluate the presence of SI. Interestingly, we discovered that females had higher rates of suicide. Referring to previous studies, contradictory results on gender as a risk factor for BD are common. Some studies showed males tended to have higher rates of completed suicide, while females had higher rates of suicide attempts [51]. Other studies identified male gender as a reliable predictor of suicidal thoughts and behaviors [52,53]. Different cultural backgrounds across nations might explain this. It is worth noting that a substantial number of the participants in this study were in their adolescence. This highlights the need for developing individual specific intervention and prevention strategies. Meanwhile, clinicians need to recognize the importance of depressive symptoms when evaluating suicide risk and address them as a potential risk factor that can be modified. Nevertheless, future therapeutic studies should explore the association between treatment effect and SI in specific subtypes of BD.

### 4.3. Strengths and Limitations

This study had some methodological strengths. First, this study involved a relatively large sample of outpatients with BD, who had no personality disorder or no substance abuse history. Previous studies suggested that substance abuse was strongly predictive of suicide in women with BD [54]. Co-morbid borderline personality disorder was also an important confounding factor [55]. Second, through an individual-centered approach, we were able to investigate symptoms in relation to how they manifest together within individuals rather than solely examining the associations between suicide and depressive or manic symptoms.

The study’s limitations also need to be considered. First, all the participants were outpatients from the Department of Psychiatry, which may not be representative of the broader BD clinical population. The three classes divided by LPA lacked an asymptomatic group and a mania group, which limited the possibility to identify unusual subtypes of BD. Second, while the sample size appeared adequate for the BD cohort, it was insufficient for LPA [56]. Collaborating to create extensive and diverse data sets could be valuable for all BD researchers. Third, the results only showed the association between depression and suicide. We were unable to ferret out the concrete factors that contribute to the relationship, such as depressed affect, poverty, functional impairment and so on. Hopelessness was thought as an independent factor associated with suicidal behavior [15]. Fourth, the current design was a single-center cross-sectional design, which limited the possibility for examining for longitudinal changes. 

## 5. Conclusions

This study pioneers the use of an individual-centered approach to examine the subtypes of BD linked to suicidal thoughts in Chinese individuals. We confirmed the idea that manic symptoms and depressive symptoms exerted no additive effects, and depressive symptoms were a strongly independent risk factor for SI. Regardless of manic symptoms, those with severe depressive symptoms had higher levels of SI. 

## Figures and Tables

**Figure 1 behavsci-14-00360-f001:**
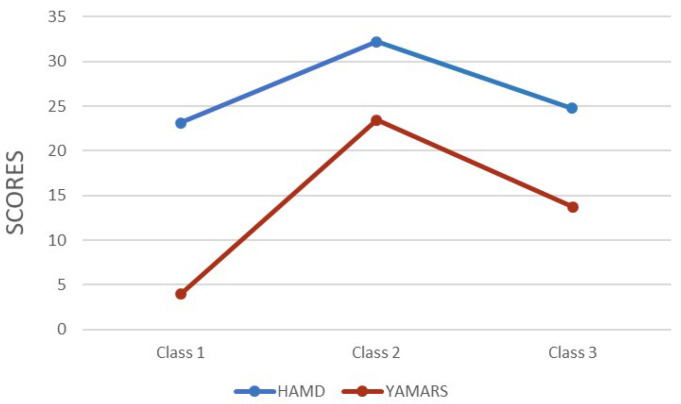
Means of HAMD-24 and YAMRS scores across three classes. Abbreviations: HAMD-24, 24-item Hamilton Depression Rating Scale; YMRS, Young Mania Rating Scale.

**Figure 2 behavsci-14-00360-f002:**
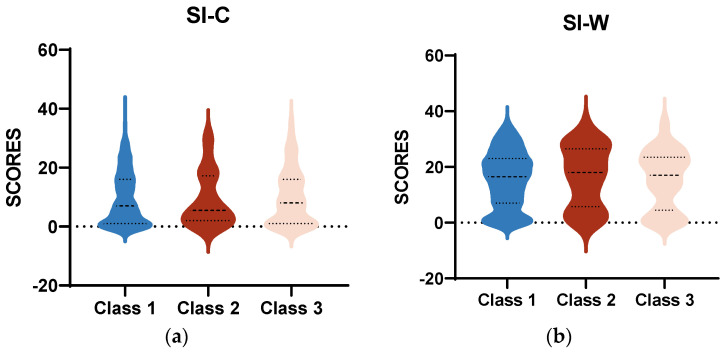
(**a**) Means of SI-C scores across three classes; (**b**) means of SI-W scores across three classes. These were conducted by one-way ANOVA. Abbreviations: SI-C, suicidal ideation-current; SI-W, suicidal ideation-worst.

**Figure 3 behavsci-14-00360-f003:**
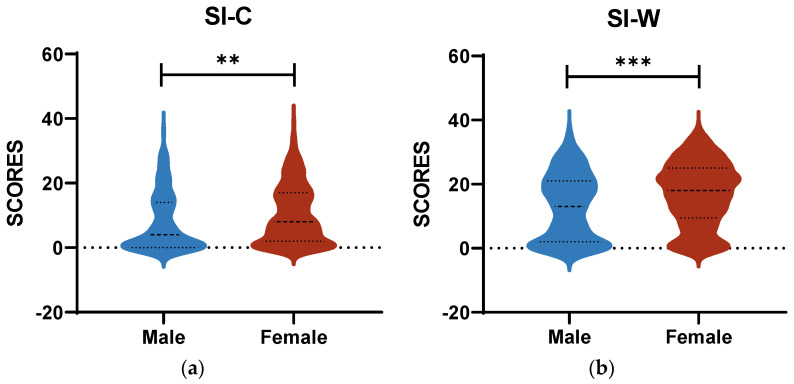
(**a**) Means of SI-C scores across different genders; (**b**) means of SI-W scores across different genders. These were conducted by independent sample *t*-tests., ** *p* < 0.01, *** *p* < 0.001.

**Table 1 behavsci-14-00360-t001:** Model fit indices of one- to six-class model and distribution.

Model	Log-Likelihood	AIC	BIC	aBIC	Entropy	Smallest Class%	LMR *p*-Value	BLRT *p*-Value
1	−3581.24	7170.49	7187.48	7174.78				
2	−3510.09	7034.18	7063.91	7041.69	0.86	17.02	0.0033	<0.0001
3	−3479.10	6978.20	7020.68	6988.94	0.88	5.80	0.0034	<0.0001
4	−3474.28	6974.57	7029.79	6988.53	0.70	5.80	0.3551	0.1111
5	−3462.37	6956.73	7024.70	6973.92	0.75	1.55	0.0333	0.0128
6	−3456.43	6950.86	7031.58	6971.27	0.79	1.74	0.8293	1.0000

Abbreviations: AIC, Akaike information criterion; BIC, Bayesian information criterion; aBIC, sample size-adjusted BIC; LMR, Lo–Mendell–Rubin-adjusted likelihood ratio test; BLRT, parametric bootstrapped likelihood ratio test.

**Table 2 behavsci-14-00360-t002:** Demographic and clinical characteristics across three classes.

Characteristics	Class 1	Class 2	Class 3	Total	*p* Value
Sample size	382 (73.89%)	30 (5.80%)	105 (20.31%)	517	
Age (year)	21.76 ± 7.31	23.17 ± 7.71	23.04 ± 8.58	22.10 ± 7.61	0.232
Gender					0.125
Female	266 (51.45%)	23 (4.45%)	64 (12.38%)	354 (68.47%)	
Male	116 (22.44%)	7 (1.35%)	41 (7.93%)	163 (31.53%)	
Height (cm)	166.32 ± 8.37	165.8 ± 7.95	166.69 ± 9.06	166.37 ± 8.48	0.864
Weight (kg)	59.76 ± 13.47	58.73 ± 13.75	61.03 ± 13.64	59.95 ± 13.50	0.619
Marriage					0.233
Yes	58 (11.22%)	3 (0.58%)	19 (3.68%)	80 (15.47%)	
No	324 (62.67%)	27 (5.22%)	86 (16.63%)	422 (81.62%)	
Education (year)	12.59 ± 3.34	14.07 ± 4.37	12.75 ± 3.37	12.71 ± 3.42	0.080
Family history					0.194
Yes	56 (10.83%)	4 (0.77%)	23 (4.25%)	83 (16.05%)	
No	326 (63.06%)	26 (5.03%)	82 (15.86%)	434 (83.95%)	
First episode					0.681
Depression	251 (48.55%)	16 (3.09%)	57 (11.03%)	324 (62.67%)	
Mania/hypomania	22 (4.26%)	1 (0.19%)	4 (0.77%)	27 (5.22%)	
Unknown	109 (21.08%)	13 (2.51%)	44 (8.51%)	166 (32.11%)	
Disease course (year)	4.41 ± 4.69	4.83 ± 4.20	4.21 ± 4.09	4.39 ± 4.55	0.818

**Table 3 behavsci-14-00360-t003:** Tests of between-subjects effects.

Source	Dependent Variable	Type III Sum of Squares	df	Mean Square	*F*	*p* Value
Corrected Model	SI-C	589.283	5	117.857	1.482	0.194
	SI-W	1006.027	5	201.205	2.123	0.062
Intercept	SI-C	147.791	1	147.791	1.859	0.173
	SI-W	1243.187	1	1243.187	13.115	0
Class	SI-C	227.435	2	113.718	1.43	0.24
	SI-W	50.579	2	25.29	0.267	0.766
Gender	SI-C	451.181	1	451.181	5.674	0.018
	SI-W	455.674	1	455.674	4.807	0.029
Class × Gender	SI-C	194.033	2	97.017	1.22	0.296
	SI-W	53.939	2	26.97	0.285	0.753
Error	SI-C	38,644.39	486	79.515		
	SI-W	46,068.37	486	94.791		
Total	SI-C	86,454	492			
	SI-W	179,147	492			
Corrected Total	SI-C	39,233.68	491			
	SI-W	47,074.4	491			

## Data Availability

All data included in this study are available upon request by contact with the corresponding author.

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
