# Peer review of "Latent Profile Analysis of Suicidal Ideation in Chinese Individuals with Bipolar Disorder"

_behavsci, 2024, doi:10.3390/bs14050360_

Round 1

Reviewer 1 Report

Comments and Suggestions for Authors

Thank the authors for the effort and the well-done wok. It's importnat to clarificate everything related to suicide and its risk factor and protective factors being this study intersting for it. Despite this, there are a few things to consider:

1.- in abstract, line 26 the acronym you use is AMD, I believe you wanted to use HAMD, for the Hamilton despression rating scale. 

2.- In the discussion lines 221-222, you must have a reference for the affirmation that manic impulsivity may elevate suicide attempt risk.

3.- Also in discussion lines 234-235, you could find lots of evidence in bibliography, specially in adolescents where female gender has higher rates of suicide attempts and suicide behaviours while male gender has more completed suicide. 

4.- In methodology you must include the study design: single-center cross-sectional study.  

Author Response

Comment 1: in abstract, line 26 the acronym you use is AMD, I believe you wanted to use HAMD, for the Hamilton despression rating scale.

Answer: Thank you for your comment. As recommended, we have revised the manuscript.

Comment 2: In the discussion lines 221-222, you must have a reference for the affirmation that manic impulsivity may elevate suicide attempt risk.

Answer: Thank you for your comment. In the revised manuscript, we have added references.

Comment 3: Also in discussion lines 234-235, you could find lots of evidence in bibliography, specially in adolescents where female gender has higher rates of suicide attempts and suicide behaviours while male gender has more completed suicide.

Answer: Thank you for your suggestion. We totally agree with you that there are lots of evidence that female gender has higher rates of suicide attempts and suicide behaviors. The literature we cited shows a higher proportion of men (14.61% vs. 10.35%) with suicidal thoughts and behaviors compared to females. However, conflicting results for gender as a risk factor in bipolar disorder and related suicide behaviors are common. As you suggested, this part has been revised.

“Referring to previous studies, contradictory results on gender as a risk factor for BD are common. Some studies showed males tended to have higher rates of completed suicide, while females had higher rates of suicide attempts. Other studies identified male gender as a reliable predictor of suicidal thoughts and behaviors.”

Comment 4: In methodology you must include the study design: single-center cross-sectional study.

Answer: Thank you for your comment. We have added the study design at the section of Materials and Methods.

Reviewer 2 Report

Comments and Suggestions for Authors

a good and decent effort on a specific population regarding the suicide in BD.

The paper has some relevance and interest for the readers.

Introduction is well constructed and sets the scene quite well.
Methods are ok , but please add the ethical no approval in the subjects subchaper.

Results are, as said, of some inters for the readers and they are fairly presented.

Discussion is I think the best part of this manuscript and brings good literature from the background which is well compared to the actual results

A fair and nice chapter of limitations.

Conclusions are short and fair. Please don t end with what should be done further (that part can be in the discussion section), but with your actual original results./

Comments on the Quality of English Language

ok

Author Response

a good and decent effort on a specific population regarding the suicide in BD.

The paper has some relevance and interest for the readers.

Introduction is well constructed and sets the scene quite well.

Methods are ok , but please add the ethical no approval in the subjects subchaper.

Answer: Thank you for your comment. We have added the ethics approval number in the subjects subchaper.

Results are, as said, of some inters for the readers and they are fairly presented.

Discussion is I think the best part of this manuscript and brings good literature from the background which is well compared to the actual results

A fair and nice chapter of limitations.

Conclusions are short and fair. Please don t end with what should be done further (that part can be in the discussion section), but with your actual original results.

Answer: Thanks for your comment. We have moved the last sentence in the Conclusion section to the Discussion section.

Reviewer 3 Report

Comments and Suggestions for Authors

A BRIEF SUMMARY 

This review article, entitled "Latent profile analysis of suicidal ideation in Chinese individuals with bipolar disorder", focuses on a pertinent topic. 

The manuscript presents an interesting and always current topic in the field of clinical psychology. The literature review is sufficient to frame the problem and to describe the state of the art of the subject under study, although it could still be improved (for example, making the LPA analyses a little more explicit). The objectives seem clear and the final results seem to contribute to a better characterisation of suicidal behaviour in clinical samples diagnosed with bipolar disorders.

Overall, the manuscript is carefully and rigorously written. The introduction is geared towards the objectives. The methodology is appropriate to the objectives. The discussion could be improved by reflecting more on the results and problematising the contrary results of previous studies. 

Title

- The title is appropriate and representative of the main goal of the study.

Key words

- The key words adequately systematise the main aspects and themes addressed in the manuscript. Suggestion: List the key words in alphabetical order. 

Abstract

- The abstract summarizes the manuscript.

I. Introduction

It is not always clear whether the results reviewed refer to Chinese samples (as in this study) or samples from other countries and cultures. 

Latent Profile Analysis (LPA): This approach could be explained a little more in the introduction, also justifying its use.

2. Materials and Methods

- Statistical analysis: This topic with the justification and description of the analysis decisions could perhaps be more condensed. 

3. Results 

- The data analysis and results reported seek to fulfil the objectives. 

- The notes in table 2 and table 3 are missing.

4. Discussion

- The discussion is presented with subtopics that make it easier to read, but there could be a more integrated articulation at the end between the objectives and the main results found in this study.

5. Conclusions

- The conclusion is clear.

References

References are adequate and up-to-date.

Check some references indicated in the text that are not included in the final references. e.g. APA (2013). DSM 5

Comments on the Quality of English Language

Nothing to report. 

Author Response

Comment 1: List the key words in alphabetical order.

Answer: Thank you for your comment. The key words have been listed in alphabetical order.

Comment 2: Latent Profile Analysis (LPA): This approach could be explained a little more in the introduction, also justifying its use.

Answer: Thanks for your comment. LPA is a widely used person-centered analysis. We have discussed the strengths of LPA.

“Latent Profile Analysis (LPA) is different from variable-centered analysis. LPA is a person-centered analysis, which uses maximum likelihood estimation to find patterns of multiple variables within individuals but not the effects of individual variables. This method enables the division of individuals into smaller and more similar subgroups.

Comment 3: The notes in table 2 and table 3 are missing.

Answer: Thanks for your suggestion. But there are no further information which needs to be noted.